# NEURAL SCALING LAWS FOR BOOSTED JET TAGGING

**Matthias Vigl**[1]     **Nicole Hartman**[1]     **Michael Kagan**[2]     **Lukas Heinrich**[1]

[1]Technical University of Munich     [2]SLAC National Accelerator Laboratory

matthias.vigl@tum.de   nicole.hartman@tum.de
makagan@slac.stanford.edu   lukas.heinrich@tum.de

## ABSTRACT

The success of Large Language Models (LLMs) has established that scaling compute, through joint increases in model capacity and dataset size, is the primary driver of performance in modern machine learning. While machine learning has long been an integral component of High Energy Physics (HEP) data analysis workflows, the compute used to train state-of-the-art HEP models remains orders of magnitude below that of industry foundation models. With scaling laws only beginning to be studied in the field, we investigate neural scaling laws for boosted jet classification using the public JetClass dataset. We derive compute optimal scaling laws and identify an effective performance limit that can be consistently approached through increased compute. We study how data repetition, common in HEP where simulation is expensive, modifies the scaling yielding a quantifiable effective dataset size gain. We then study how the scaling coefficients and asymptotic performance limits vary with the choice of input features and particle multiplicity, demonstrating that increased compute reliably drives performance toward an asymptotic limit, and that more expressive, lower-level features can raise the performance limit and improve results at fixed dataset size.

## 1   INTRODUCTION

Machine learning has become a central tool in High-Energy Physics (HEP) data analysis, with deep neural networks now routinely deployed for tasks such as jet tagging, event classification, reconstruction and anomaly detection. In proton–proton collisions at the Large Hadron Collider (LHC) (Evans & Bryant (2008)), quarks and gluons produced in hard scattering processes cannot be observed directly; instead, they fragment and hadronize into collimated sprays of particles known as jets. When a heavy particle such as a top quark, Higgs boson, or $W/Z$ boson is produced with transverse momentum much larger than its mass, its decay products merge into a single large-radius "boosted" jet whose internal radiation pattern, or substructure, encodes the identity of the parent particle. Classifying these boosted jets, that is distinguishing for example a top-quark jet from a generic quark- or gluon-initiated jet, is a key task at the LHC and a succession of increasingly expressive architectures has driven steady improvements in this discrimination. Yet despite this progress, the compute budgets used to train state-of-the-art jet tagging models (ATLAS (2025; 2026a); CMS (2025); Qu & Gouskos (2020); Kasieczka et al. (2019); Gong et al. (2022); Bogatskiy et al. (2024); Qu et al. (2024); Brehmer et al. (2025); Mikuni & Nachman (2025a;b); Smith et al. (2023)) remain orders of magnitude below those of industry foundation models. In parallel, the success of Large Language Models has established a powerful empirical principle: scaling compute, through joint increases in model capacity and training data, is the primary driver of performance gains in modern deep learning. The resulting *neural scaling laws*, formalized by Kaplan et al. (2020) and refined into compute-optimal prescriptions by Hoffmann et al. (2022), provide a quantitative framework for predicting how loss decreases as a power law in compute and for allocating resources efficiently between model size and data. These insights have reshaped how models are developed across natural language processing (OpenAI (2024)) and computer vision (Zhai et al. (2022)), but their applicability to scientific domains with distinct data generation processes and task structures remains an open question. Furthermore, the emergence of foundation models in HEP (Amram et al. (2025); Leigh et al. (2024); Golling et al. (2024); Vigl et al. (2024); Zhao et al. (2024); Birk et al. (2024; 2025a;b); Giroux & Fanelli (2025); Bhimji et al. (2025); Hsu et al. (2026); Park et al. (2025)) has driven a trend

toward larger architectures trained on diverse, multi-modal inputs: understanding how to effectively scale these models across data, parameters, and compute, is essential for realizing their full potential in HEP.

## 1.1 RELATED WORK

There is a growing effort within the HEP community to understand in detail the fundamental limits of jet tagging performance, with different approaches being explored (Amram et al. (2025); Geuskens et al. (2025); Pang et al. (2025)). On the scaling laws side, Batson & Kahn (2023) demonstrated the emergence of power-law scaling of the test loss as a function of training set size for several physically-motivated classifiers in the top-vs-QCD jet discrimination task, showing that different classifiers exhibit distinct scaling exponents and that their relative ranking can change as the dataset grows. More recently, ATLAS (2026b) conducted a large-scale study of transformer scaling behavior for heavy-flavor jet identification using up to $10^{10}$ simulated jets, and jointly scaled dataset size and model size up to 3 billion parameters. Park et al. (2025) demonstrated power-law scaling of the pretraining loss of foundation model for nuclear and particle physics with respect to model size, dataset size, and compute, and Bahl et al. (2026) derived scaling laws for amplitude surrogate models.

In this work, we approach the question of fundamental limits of jet tagging performance through the lens of neural scaling laws. Using the public JetClass dataset (Qu et al. (2022)), we systematically vary model size and training dataset size to derive compute-optimal scaling relations for boosted jet classification. We identify an irreducible loss representing the asymptotic performance limit, study how data repetition modifies the scaling behavior, and investigate how the choice of input features and particle multiplicity shifts the performance ceiling. Our results establish that scaling laws provide a predictive framework for understanding the limits of jet tagging and for guiding resource allocation in future HEP machine learning efforts.

## 2 DATASET AND TRAINING SETUP

We describe here the dataset, model architecture, and training procedure used throughout this work.

**Dataset.** JetClass (Qu et al. (2022)): This dataset contains 100M simulated jets for training, 5M for validation and 20M for testing. Jets contain up to 128 particles and are divided into 10 classes, with QCD jets being the *background* class. The production and decay of top quarks and of the $W$, $Z$, and Higgs bosons are simulated using MADGRAPH5_AMC@NLO (Alwall et al. (2014)). The subsequent evolution of the generated events, including parton showering and hadronization, is modeled with PYTHIA (Sjöstrand et al. (2015)), yielding the final-state particles. The per-particle inputs include kinematic variables (momentum, energy, and angular coordinates), particle-type labels (leptons, photons, and charged or neutral hadrons) assigned by the detector's particle flow reconstruction, and variables characterizing the displacement of each particle's trajectory from the collision point: specifically, the signed transverse and longitudinal impact parameters and their uncertainties, which help identify particles originating from displaced decays of long-lived hadrons.

**Model.** All experiments are based on a set Transformer encoder (Vaswani et al. (2023); Lee et al. (2019)) architecture. Jets are represented as variable-length sequences of up to $N$ constituent particles. No positional encoding is applied, so the architecture is invariant to the ordering of constituents. We sort particles by decreasing transverse momentum $p_T$ only to define a deterministic truncation policy when varying the number of particles considered. Each particle is described by a feature vector comprising kinematic variables, particle identification flags and track parameters, for a total of 21 features. A linear projection maps each particle feature vector to a $d$-dimensional embedding space. A learnable class token `[CLS]` (Devlin et al. (2019)) is prepended to the sequence, and the resulting $(N+1)$-token sequence is processed by a Transformer encoder with 4 layers. Each layer applies pre-layer-normalization (Xiong et al. (2020)), multi-head self-attention with $h = \max(1, \lfloor d/64 \rfloor)$ heads, and a feed-forward network with hidden dimension $4d$ and GELU activation (Hendrycks & Gimpel (2023)), with dropout (Srivastava et al. (2014)) applied at rate $p = 0.01$. Zero-padded particle positions are excluded from attention via a key-padding mask. The encoder output corresponding to the class token serves as a fixed-size summary of the entire jet; this representation is layer-normalized and linearly projected to produce logits over the 10 jet classes.

**Compute.** Compute cost for training such a model of size $N$ parameters on $D$ samples can be approximated by $C = n_p 6ND$ FLOPs per epoch (Kaplan et al. (2020)), where the factor of 6 accounts for the two multiplications per parameter in the forward pass and the roughly $2\times$ cost of the backward pass, while $n_p$ denotes the particle multiplicity, or number of tokens, per jet: on average $n_p \approx 40$.

**Training.** All models are trained with a batch size of 128, AdamW (Loshchilov & Hutter (2019)) optimizer (learning rate $= 10^{-4}$, weight decay $= 10^{-2}$) and standard cross-entropy loss. Model capacity (number of parameters) is scaled by varying the embedding dimension.

## 3 SCALING LAWS

In this section, we study compute optimal neural scaling laws for jet classification by systematically varying the model capacity $N$ and the training dataset size $D$. In the strictly compute optimal training regime, data is not repeated during training to maximize the efficiency of each update step. In HEP and many scientific applications, however, models are typically trained for multiple epochs over the same dataset. We therefore also examine how data repetition affects the scaling exponents and the additional compute required to achieve a given performance. Finally, we investigate how the scaling behavior depends on the choice of per-particle input features and on the particle multiplicity used to represent each jet.

### 3.1 COMPUTE-OPTIMAL SCALING

As as introduced for LLMs by Hoffmann et al. (2022), we model the Loss as a function of model size $N$ and dataset size $D$ using the parametric form:

$$L(N, D) = L_\infty + \frac{A}{N^\alpha} + \frac{B}{D^\beta}, \tag{1}$$

where $L_\infty$ represents the irreducible loss, the best achievable performance in the limit of infinite model size and data, $A/N^\alpha$ captures the contribution from finite model capacity, and $B/D^\beta$ captures the contribution from finite dataset size. The exponents $\alpha$ and $\beta$ govern how rapidly each source of error diminishes with scale. We find good agreement with this functional form with fitted parameters shown in Table 1.

The compute-optimal allocation correspond to the $(N, D)$ configuration that minimizes $L$ for a given compute budget $C$, i.e:

$$\min_{N,D} L(N, D) \quad \text{s.t.} \quad C = n_p 6ND = C_{\text{budget}}, \tag{2}$$

where the total training compute $C = n_p 6ND$ follows from the per-sample cost of a Transformer forward and backward pass (Section 2). Solving (2) yields the optimal scalings:

$$N \propto C^a, \qquad D \propto C^{1-a}, \qquad L \propto C^{-\gamma}, \qquad \text{with } a = \frac{\beta}{\alpha + \beta}. \tag{3}$$

Table 1: Scaling-law parameters from fits to $L_\infty + A/N^\alpha + B/D^\beta$. Under compute-optimal scaling, Loss follows $L \propto C^\gamma$. Reported uncertainties are 68% bootstrap confidence intervals. In order to reduce uncertainties in $L_\infty$, its value is constrained to that obtained in Section 3.2, which probes significantly higher compute regions. For reference, Hoffmann et al. (2022) found $\alpha \approx 0.38$ and $\beta \approx 0.28$ for LLMs.

| Parameter | Value (68% CI) |
|---|---|
| $L_\infty$ | 0.32 (0.29, 0.34) |
| $A$ | 11.27 (8.01, 15.14) |
| $\alpha$ | 0.44 (0.40, 0.48) |
| $B$ | 7.22 (6.66, 7.89) |
| $\beta$ | 0.22 (0.21, 0.23) |
| $\gamma$ | 0.15 (0.13, 0.16) |

The resulting compute-optimal scaling trajectory is shown as the blue dashed line in Fig. 1. In practice, this trajectory can be followed only up to relatively modest compute budgets, corresponding to the regime in which the full training dataset is seen once. Typical jet datasets in high-energy physics contain a few $10^8$ jets, with recent efforts pushing up to $10^{10}$ (ATLAS (2026b)), beyond which additional compute is necessarily spent on training for multiple epochs over the same data.

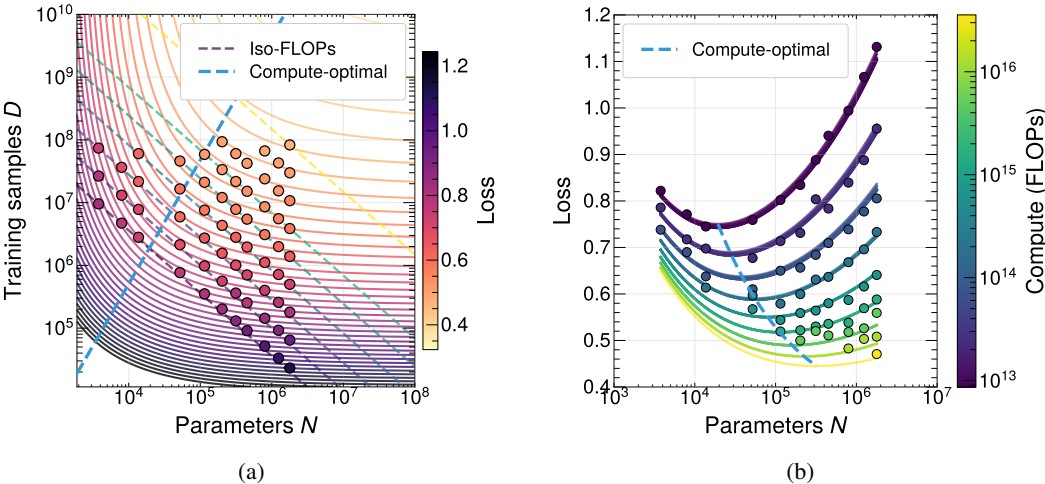

(a)                                   (b)

Figure 1: Each point represents the validation loss of a Transformer encoder with $N$ parameters (varying the embedding dimension) trained on $D$ unique samples for exactly one epoch (no data repetition). Models are trained across a grid of $(N,D)$ configurations spanning several orders of magnitude in both axes. The parametric form in Eq. (1) is then fit jointly to all points. (a) Loss surface $L(N, D)$ as a function of model size $N$ and training dataset size $D$, with iso-loss contours shown as colored lines and the compute-optimal trajectory as the blue dashed line. The iso-FLOP lines are colored by the corresponding compute budget, with the color scale shown in (b). (b) Loss as a function of model size at fixed compute budget (iso-FLOP curves), with the compute-optimal trajectory crossing the minima of each curve.

## 3.2 SCALING UNDER DATA REPETITION

Even though an unlimited number of jets could in principle be generated, producing simulation data is computationally expensive and thus dataset size is often fixed in practice. In this setting, multiple epochs can still improve performance beyond what a single pass achieves; update steps however become less compute optimal over repeated passes causing the validation loss to saturate or eventually overfit for large enough models. This simultaneously increases the total training compute required to achieve a given performance relative to the unconstrained data regime, as shown in Fig. 2a, where models trained with data repetition always lie above the compute optimal line derived before.

When training with data repetition, the goal is to exploit excess compute to extract maximal performance from a fixed dataset. In this regime, models must be sufficiently large to avoid the underparameterized regime and reach the minimum achievable validation loss; in practice, this requires choosing model sizes above an overfitting threshold $N \propto D^\lambda$. To determine the overfitting threshold, we train models across a grid of $(N, D)$ configurations and classify each as underfitting or overfitting based on whether the validation loss plateaus or begins to increase with further training. The resulting phase diagram is shown in Fig. 3. A power-law fit yields the overfitting threshold $N \propto D^{0.47}$, indicating a roughly square-root scaling between the minimum model size required to overfit and the dataset size. For model sizes exceeding this threshold, the minimum validation loss lands at approximately the same value, such that multiple choices of $N > D^\lambda$ yield comparable performance when trained until overfitting on the fixed dataset $D$. This is illustrated in Fig. 2a, where for a fixed dataset size, all models with enough capacity to overfit converge to similar loss values, regardless of further increases in $N$. Models near the overfitting threshold are however preferred, as they remain close to compute-efficient and relatively small in size.

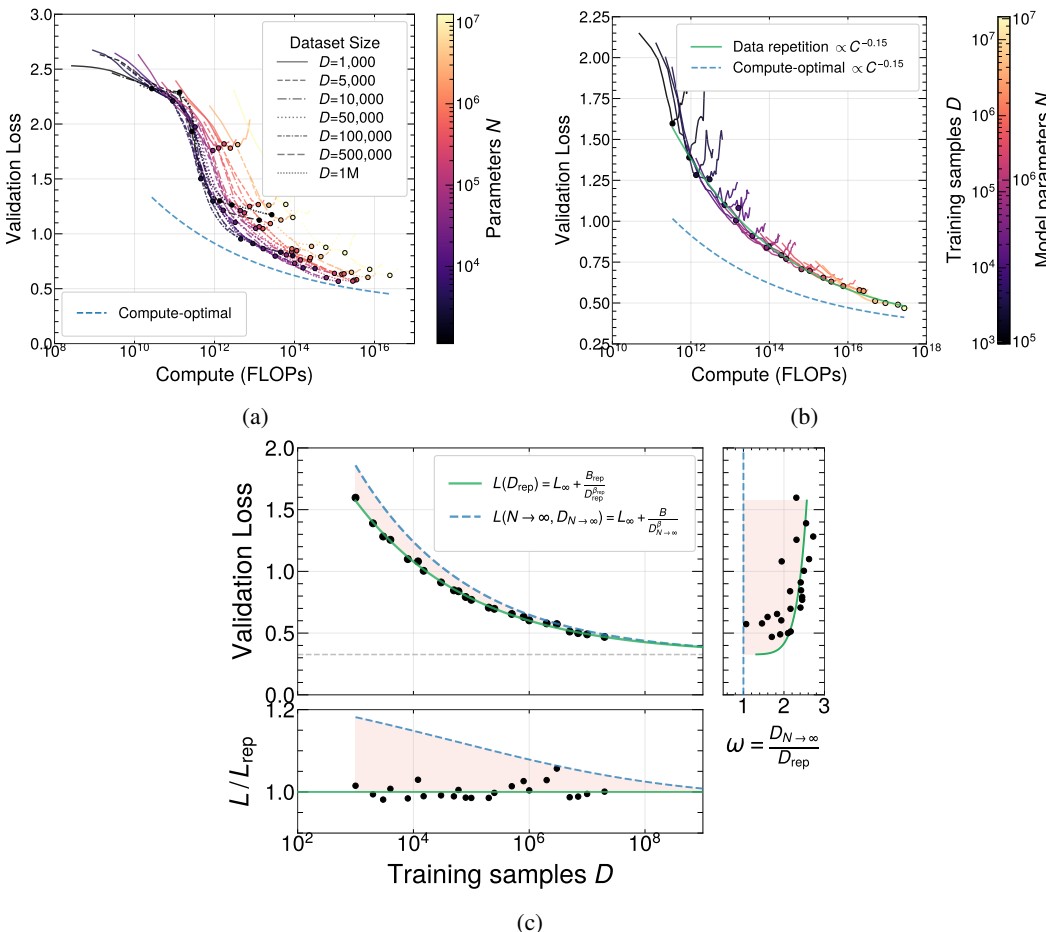

Figure 2: Scaling under data repetition. (a) Validation loss as a function of total training compute for models trained on fixed dataset sizes ranging from D = 1k to D = 1M, compared to the compute-optimal scaling $L \propto C^{-0.15}$ (dashed line). Each point represents the early-stopped validation loss of a single model, obtained by stopping training at the epoch with lowest validation loss. Models are trained across a two-dimensional grid: the dataset size $D$ is indicated by the line style, with the embedding dimension $d \in \{4, 8, 16, 32, 64, 96, 128, 256, 512\}$ also being varied and the resulting model size $N$ indicated by the color scale. Models trained with data repetition consistently lie above the compute-optimal frontier, with validation loss eventually either saturating or overfitting. (b) Training model sizes above the overfitting threshold allows to minimize the validation loss at each fixed dataset size. Scaling along this trajectory yields the same compute exponent, at the price of roughly a factor of 10 in compute to reach the same loss as compute-optimal scaling with no data repetition. (c) Early stopped validation loss as a function of dataset size $D$ for models trained above the overfitting threshold with data repetition (black points), compared to scaling with no data repetition under the large $N$ regime (negligible model size term) $L(N \to \infty, D) = L_\infty + B/D^\beta$. Both curves are compute sub-optimal, as they correspond to the limit of large model size rather than the compute-optimal allocation. Although training on repeated data effectively amplifies the dataset, the gain is bounded by $\omega D_{\mathrm{rep}}$.

By training models above the overfitting threshold in Fig. 2b, we find that while the the scaling exponent $\beta_{\mathrm{rep}} \sim \beta$ remains approximately unchanged, data repetition primarily modifies the $B_{\mathrm{rep}}$ normalization of a scaling law of the type:

$$L(D_{\mathrm{rep}}) = L_\infty + B_{\mathrm{rep}}/D_{\mathrm{rep}}^{\beta_{\mathrm{rep}}}, \qquad (4)$$

corresponding to an improvement in data efficiency at expense of additional compute. Notably, the $N$ dependence effectively drops out in the repeated-data regime: once model size is above

the overfitting threshold, increasing $N$ no longer reduces the converged loss. Second-order effects such as double descent could in principle yield additional gains from increasing $N$ beyond the interpolation threshold (Nakkiran et al. (2019); Vigl & Heinrich (2025)); however, this effect was found to be at the percent level in loss by ATLAS (2026b). At sufficiently large compute the benefits of data repetition saturate, leading to a plateau or eventual degradation of validation performance due to overfitting. As a result, increasing compute through repeated epochs becomes progressively less efficient than scaling the dataset size. Setting Eq. (4) equal to Eq. (1) in the limit of large model size ($N \to \infty$), we can introduce a notion of *effective size* $D_{N \to \infty} = \omega D_{\text{rep}}$ of a fixed dataset $D_{\text{rep}}$, where $\omega$ is the amplification factor such that training on $D_{\text{rep}}$ samples for multiple epochs until overfitting yields a loss equivalent to a single pass over $D_{N \to \infty}$, with:

$$\omega(D_{\text{rep}}) = \left( \frac{B_{\text{no-rep}}}{B_{\text{rep}}} \right)^{1/\beta_{\text{no-rep}}} \cdot D_{\text{rep}}^{\beta_{\text{rep}}/\beta_{\text{no-rep}} - 1}. \tag{5}$$

This quantity captures the diminishing returns of repetition: although training on repeated data effectively amplifies the dataset, the gain is bounded by $\omega D_{\text{rep}}$, as illustrated in Fig. 2c. Consequently, data repetition is preferable to generating new simulation data only when the effective gain exceeds what could be achieved with equivalent computational resources devoted to simulation instead. Beyond this factor, and with no modification to the architecture beyond size, further improvements require generating additional simulation data. It is important to note that this interpretation of the measured effective size is only viable under the assumption that Eq. 1 holds for $N \to \infty$ (blue dashed line in Fig. 2c). Muennighoff et al. (2025) explore a parametrization where excess model parameters and repeated passes lead to an exponential decay of their respective terms in Eq. 1. This parametrization however doesn't allow for excess parameters or excess epochs to negatively impact performance making it hard to fit to our experimental results, as loss can start to increase even for few repeated passes, as it can be seen in Fig. 2b, with smoothness not recovered by epoch-wise double descent for a wide range of model sizes (ATLAS (2026b)). For this reason we decided to only model the early stopped validation loss past overfitting threshold, which exhibits a smooth power law scaling in $D$ and it's independent from model size $N$.

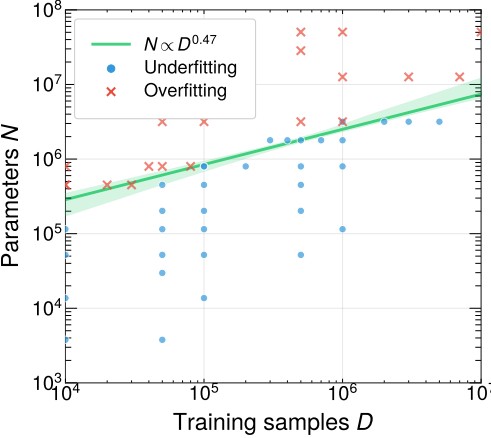

Figure 3: Overfitting threshold in the $(N, D)$ plane. Blue dots indicate models in the underfitting regime, where increasing training time leads to a plateau in the validation loss. Red crosses indicate models that overfit, where the validation loss eventually increases with further training. The green line shows the fitted threshold $N \propto D^{0.47}$, corresponding to a roughly square-root scaling between the minimum model size needed to overfit and the dataset size. 68% confidence intervals are obtained by bootstrapping threshold points.

## 3.3 INPUT FEATURES DEPENDENCE

The scaling laws derived in Sections 3 and 3.2 were obtained using the full set of 21 per-particle input features and up to 128 particles per jet. To investigate how the choice of input representation affects scaling behavior, we train models above the overfitting threshold for four configurations.

**Input configurations.** We consider four configurations of increasing expressiveness: (i) kinematic variables only ($\Delta\eta$, $\Delta\phi$, $\log p_T$) with 40 particles, (ii) the full 21-feature set with 10 particles, (iii) the full 21-feature set with 40 particles, and (iv) the full 21-feature set with 128 particles per jet. In all cases, particles are ordered by decreasing $p_T$, such that configurations with fewer particles retain only the hardest constituents. For each configuration, we vary the training dataset size $D$ and fit the scaling form (4).

Results are shown in Fig. 4 and summarized in Table 2. The data scaling exponent $\beta_{\text{rep}}$ remains approximately constant across all configurations, ranging from 0.21 to 0.26, indicating that the rate at which additional data reduces the loss is largely independent of the input representation. In contrast, the asymptotic loss $L_\infty$ varies significantly. The small difference between 40 and 128 particles suggests that most of the physics-relevant information is captured by the leading $\sim$40 constituents, consistent with the average particle multiplicity in the dataset.

These results demonstrate that the choice of input features primarily affects the irreducible loss $L_\infty$, that is, the asymptotic performance ceiling, rather than the scaling rate. More expressive, lower-level features lower this ceiling, enabling better performance not only in the infinite-data limit but also at any fixed dataset size.

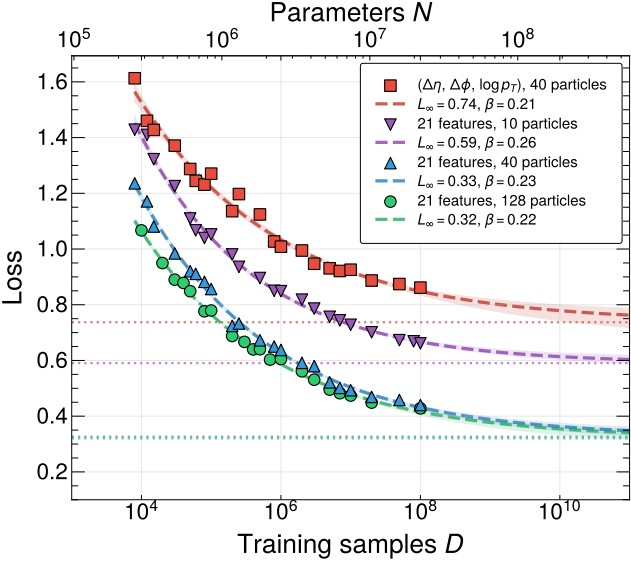

Figure 4: Loss as a function of training dataset size $D$ for models trained above the overfitting threshold, using four configurations: kinematic variables only ($\Delta\eta, \Delta\phi, log\, p_T$), and the full feature set with 10, 40, and 128 particles per jet. Dashed lines show fits to $L(D_{\text{rep}}) = L_\infty + B_{\text{rep}}/D_{\text{rep}}^{\beta_{\text{rep}}}$. The scaling exponent $\beta_{\text{rep}}$ is roughly constant across configurations, while the asymptotic loss $L_\infty$ decreases with richer inputs and higher particle multiplicity, which also reach the same performance with significantly less amounts of data.

Table 2: Scaling parameters from fits to $L(D_{\text{rep}}) = L_\infty + B_{\text{rep}}/D_{\text{rep}}^{\beta_{\text{rep}}}$ for four input feature configurations. Reported uncertainties are 68% bootstrap confidence intervals.

| Configuration | $L_\infty$ (68% CI) | $B_{\text{rep}}$ (68% CI) | $\beta_{\text{rep}}$ (68% CI) |
|---|---|---|---|
| ($\Delta\eta$, $\Delta\phi$, $\log p_T$), 40 particles | 0.74 (0.67, 0.77) | 5.62 (4.16, 7.10) | 0.21 (0.17, 0.24) |
| 21 features, 10 particles | 0.59 (0.57, 0.61) | 8.69 (7.59, 10.35) | 0.26 (0.24, 0.28) |
| 21 features, 40 particles | 0.33 (0.30, 0.34) | 7.08 (6.27, 7.79) | 0.23 (0.21, 0.24) |
| 21 features, 128 particles | 0.32 (0.29, 0.34) | 5.79 (5.06, 6.27) | 0.22 (0.20, 0.23) |

## 4   PHYSICS PERFORMANCE

Having established that the parametric forms (1,4), provide a good description of the scaling behavior for jet classification, we now translate these results into physics-relevant metrics, that is QCD background jets rejection at fixed signal jets efficiency. Following ATLAS (2026b), we train models above the overfitting threshold as in Section 3.2 on increasing dataset size $D$, and translate cross-entropy loss to QCD background rejection via a fit to the empirical mapping $R(L) = Ae^{BL} + C$ in Fig. 5. Signal efficiency is fixed at 50% for all signal jet types, except for $H \rightarrow l\nu qq'$ (99%) and $t \rightarrow bl\nu$ (99.5%). Combined with the scaling laws derived in Section 3, this enables direct prediction of expected physics performance as a function of compute. The performance of the ParT (plain) architecture (as reported in Table 2 in Qu et al. (2024)) trained on the full 100M dataset is reproduced at the corresponding scale, confirming consistency between the scaling predictions and established benchmarks. Beyond this point, scaling laws predict that further increases in dataset size and compute continue to improve background rejection.

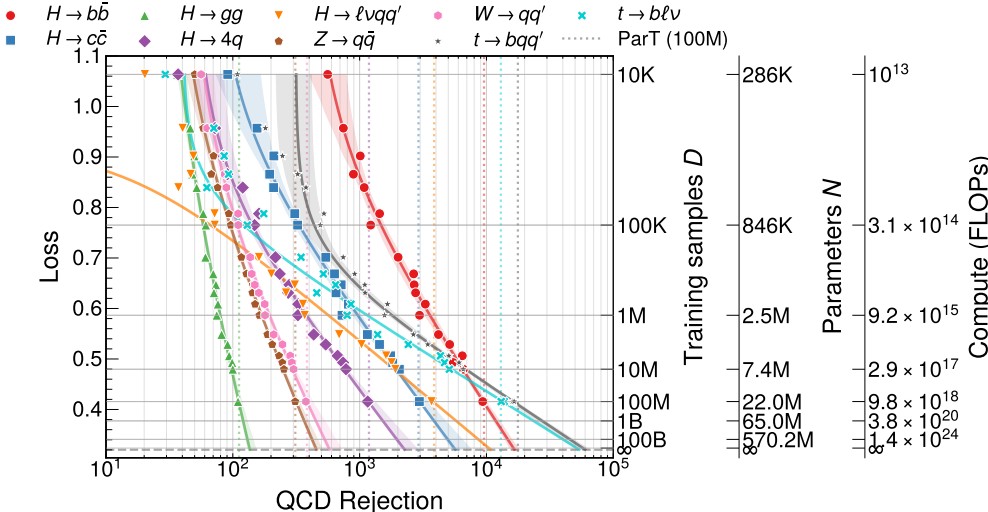

Figure 5: QCD background rejection for each signal class at 50% efficiency, except for $H \rightarrow l\nu qq'$ (99%) and $t \rightarrow bl\nu$ (99.5%), as a function of total loss and dataset size when training for multiple epochs above the overfitting threshold. The performance of the ParT architecture trained on 100M samples is shown for reference (dashed lines), which is crossed as expected at the 100M jets scale level. Each colored curve corresponds to one of the nine signal processes in the JetClass dataset.

Fig. 6a shows the ROC curves for top quark tagging across the four input feature configurations studied in Section 3.3, evaluated on models trained with $10^8$ samples. The dashed lines indicate the asymptotic performance expected in the $D \rightarrow \infty$ limit, obtained from the fitted $L_\infty$. Notably, the observed ROC curve for the $(\Delta\eta, \Delta\phi, \log p_T)$ configuration at the $D = 10^8$ scale is compatible with the JetClass OmniLearn result reported in Pang et al. (2025), with the fundamental limit predicted by scaling laws shown with a dashed line.

These results confirm that richer input representations and higher particle multiplicities yield substantially higher QCD rejection, consistent with the lower asymptotic losses observed in the scaling fits in Fig. 4. It is important to note that the loss $L$ refers to the total cross-entropy summed over all jet types: when varying input features and particle multiplicity, individual signal processes benefit differently depending on which physics information is gained or lost, as illustrated in Fig. 6b where QCD rejection at 50% top signal efficiency exhibits a different scaling exponent when limiting the number of particles to 10.

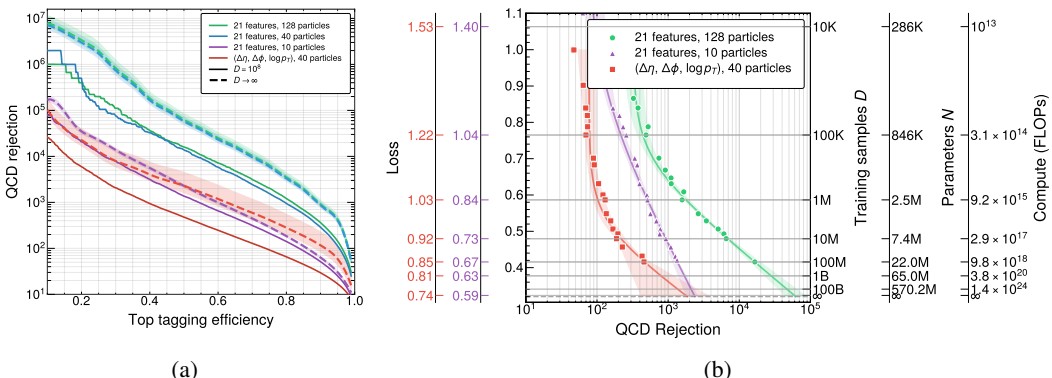

(a)              (b)

Figure 6: Top vs QCD jet tagging. (a) ROC curves for top quark tagging across the four input feature configurations, for models trained on $D = 10^8$ samples (solid lines) and the asymptotic performance in the limit $D \to \infty$ (dashed lines) obtained from the fitted $L_\infty$. (b) QCD background rejection at 50% top signal efficiency as a function of total loss and dataset size for the 128-particle, 10-particle, and kinematic-only ($\Delta\eta$, $\Delta\phi$, $\log p_T$) configurations. While all configurations follow a smooth scaling with dataset size, individual signal processes exhibit different scaling exponents depending on which physics information is retained by the input representation.

## 5 CONCLUSION

We have presented a systematic study of neural scaling laws for boosted jet classification using the public JetClass dataset. By jointly varying model capacity $N$ and training dataset size $D$, we showed that the cross-entropy loss is well described by the parametric form (1) observed in language and vision models. We identified an irreducible loss $L_\infty$ that represents the best achievable performance in the limit of infinite model size and data. This asymptotic limit depends on the choice of input representation and particle multiplicity: crucially, the data scaling exponent $\beta$ remains approximately constant across the input configurations studied here, suggesting that richer, lower-level features may raise the performance ceiling without substantially altering the rate at which data reduces the loss. A more systematic study across a broader range of input modalities would be needed to confirm the generality of this observation. We further studied the effect of data repetition, which is common practice in HEP where current dataset sizes are limited by the cost of simulation. Training for multiple epochs above the overfitting threshold effectively reduces the prefactor $B$ in the scaling law while roughly preserving $\beta$. However, this comes at a roughly tenfold increase in compute relative to the single-pass optimal regime, and the gains from repetition eventually saturate. Finally, we translated the scaling laws into physics-relevant metrics by mapping cross-entropy loss to QCD background rejection at fixed signal efficiency. Notably, the asymptotic performance limits obtained here with fast simulation saturate at lower scales than observed by ATLAS (2026b) for small radius jets tagging on full detector simulation. This suggests that simulation fidelity may itself be a limiting factor in jet tagging performance, and that scaling laws could serve as a diagnostic tool for quantifying the impact of simulation quality on achievable discrimination power. Our results establish that scaling compute drives performance toward a well-defined asymptotic limit for jet classification, and that this limit can be improved by the use of more expressive, lower-level input features. These findings motivate further investigation into scaling laws across other physics tasks and architectures. Understanding architecture-dependent scaling behaviors with dataset size and compute is essential for making informed decisions about model selection and deployment in LHC experiments.

## ACKNOWLEDGEMENTS

M.K. is supported by the US Department of Energy (DOE) under Grant No. DE-AC02-76SF00515. LH and NH are supported by the Excellence Cluster ORIGINS, which is funded by the Deutsche Forschungsgemeinschaft (DFG, German Re- search Foundation) under Germany's Excellence Strategy - EXC-2094-390783311. LH and MV are supported by BMFTR Project SciFM 05D2025.

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

# A  APPENDIX

## A.1  COMPUTE COST

The training cost of a Transformer approximately scales as $6N$ FLOPs per token, where $N$ is the number of parameters and the factor of 6 accounts for two multiply-accumulate operations per parameter in the forward pass and the roughly $2\times$ cost of the backward pass (Kaplan et al. (2020)). Figure 7 compares the FLOPs empirically measured by `torch.utils.flop_counter.Flop-CounterMode` for models used in this study to the $6Nn_p$ estimate as a function of the number of parameters, for $n_p = 40$ particles, confirming that the parametrization provides a reliable estimate of the training compute.

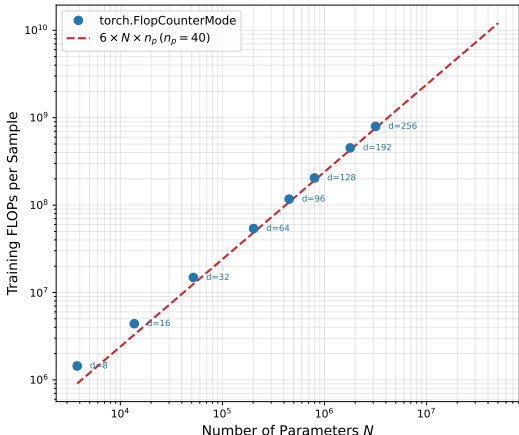

Figure 7: Training FLOPs per sample for the Transformer encoder architecture used in this work as a function of the number of parameters $N$, for $n_p = 40$ particles. Blue dots show the values measured with `torch.utils.flop_counter.FlopCounterMode` while the red dashed line shows the $6 \times N \times n_p$ approximation. The two estimates are in good agreement, validating the use of the $6Nn_p$ approximation for the compute budget.

