# OpenReview forum: "Neural Scaling Laws for Boosted Jet Tagging"
_ICLR.cc/2026/Workshop/FM4Science — ICLR 2026 Workshop FM4Science Poster_

### Official Review · Reviewer_FYzk · 2026-02-13
**Systematic Investigation of Neural Scaling Laws and Asymptotic Limits in Boosted Jet Tagging**

**Rating:** 8
**Confidence:** 4

**Review:**

**Summary**

This paper presents a rigorous empirical study of neural scaling laws within the domain of High Energy Physics (HEP), specifically focusing on boosted jet classification using the JetClass dataset. By systematically varying model capacity ($N$) and training dataset size ($D$), the authors derive compute-optimal scaling relations and identify an irreducible loss ($L_{\infty}$) that represents the fundamental performance ceiling for a given input representation. The work also quantifies the impact of data repetition—a common necessity in HEP due to expensive simulation costs—and demonstrates how feature expressiveness shifts the asymptotic performance limit without significantly altering the scaling rate.

**Originality and Significance**

While neural scaling laws are well-established in NLP and CV, their application to scientific domains with distinct data structures, such as the jet substructure in HEP, is relatively nascent. This paper provides a necessary bridge between industry-standard scaling practices and the specific constraints of physics experiments. The discovery that lower-level, more expressive features primarily lower the irreducible loss ceiling rather than changing the scaling exponent is a significant insight for future architecture and feature engineering in HEP foundation models.

**Quality and Clarity**

The methodology is sound and the paper is exceptionally clear. The authors utilize a Transformer encoder architecture and follow the compute-optimal prescriptions established by Hoffmann et al. (2022) . The use of 68% bootstrap confidence intervals for scaling parameters adds a layer of statistical rigor often missing in empirical scaling studies. The translation of cross-entropy loss into physics-relevant metrics (QCD background rejection) makes the findings actionable for experimentalists.

**Pros**

*1. Predictive Framework:* Establishes a quantitative way to allocate compute resources between model size and data for jet tagging tasks.

*2. Irreducible Loss Identification:* The identification of $L_{\infty}$ provides a "speed limit" for current tagging methods, helping researchers understand when to stop scaling and when to change input representations.

*3. Data Repetition Analysis:* Provides a realistic assessment of the "cost of simulation" by showing that data repetition requires a tenfold increase in compute to match single-pass efficiency.

*4. Feature Expressiveness:8 Clearly illustrates that richer features (e.g., 21 features vs. kinematics only) provide a more effective route to performance gains than simply increasing dataset size for less-expressive models.

**Cons**

*1. Contextualizing Model Selection:* The authors mention that scaling helps in "making informed decisions about model selection". To make this claim more robust, they should reference recent work on how fine-tuning dynamics impact this selection process (e.g., LensLLM). This would clarify if the observed scaling laws hold true after the models undergo task-specific adaptation.

*2. Architecture Specificity:* The study is limited to Transformer encoders. While these are state-of-the-art, the scaling exponents $(\alpha, \beta)$ might vary significantly for GNNs or Lorentz-equivariant architectures like ParT.

*3. Emerging HEP Foundations:* While several foundation models are cited (e.g., Evenet), the authors could better discuss how their derived scaling laws for classification specifically inform the training of these multi-modal, cross-task models.

Reference:
1. Zeng, X., et al. (2025). LENSLLM: Unveiling Fine-Tuning Dynamics for LLM Selection. arXiv:2505.03793.
2. Hsu, T. H., et al. (2026). Evenet: A foundation model for particle collision data analysis. arXiv:2601.17126.

---

### Official Review · Reviewer_tq4i · 2026-02-22
**Scaling-law study is solid, but compute accounting and training protocol weaken the main claims**

**Rating:** 4
**Confidence:** 3

**Review:**

The paper fits Kaplan/Hoffmann-style scaling laws for JetClass boosted-jet classification using a plain Transformer encoder by varying model size and dataset size, then studies data repetition and input representation effects. The empirical trends are plausible, but the compute definition, “compute-optimal” protocol (1 epoch, no convergence), and several subjective choices (overfitting threshold labeling, loss-to-physics mapping) make the quantitative conclusions less reliable than the paper implies.

You train a 4-layer Transformer encoder on JetClass, vary model size by embedding dimension, vary dataset size, and fit a standard scaling-law surface with an irreducible loss term. You then study repeated-epoch training on fixed data, define an overfitting threshold, and map loss to physics metrics like QCD rejection, with a ParT reference point.


Strengths?

The experimental grid over model size and dataset size is the right structure for a scaling-law paper.

The repetition analysis is relevant to HEP where simulation is expensive, and you quantify the compute penalty.

The feature and particle-count comparisons are useful and practically motivated.

The attempt to translate loss into physics-facing performance is a good direction.


Major concerns?

Compute is treated as proportional to parameters times tokens times samples, which is not a faithful compute model for Transformers. If you claim “compute-optimal scaling,” you need measured FLOPs, or at least a validated FLOPs model that reflects attention and MLP costs.

Your “compute-optimal” grid trains for exactly one epoch. That is a training-budget choice, not a compute-optimal result, unless you show that one epoch is near-optimal across scales. Otherwise you are fitting training dynamics, not just scaling.

Hyperparameters are fixed across large changes in model and data scale. Scaling exponents can change when training is tuned per scale. You do not show robustness.

The overfitting threshold depends on a subjective labeling rule (validation loss “plateaus” vs “increases with further training”). This needs an objective rule and a sensitivity analysis.

The “effective dataset size” interpretation under repetition is explicitly conditional on assumptions. That is fine, but then the paper should be more cautious with quantitative claims.

The loss to QCD rejection mapping is fit empirically. You need confidence bands and evidence it holds across model sizes and input settings, not just one setting.

Sloppy citations and metadata reduce trust (example: “Vaswani et al. (2023)” and the “Eva (2008)” entry).


Required changes for acceptance?

Re-do the compute story using measured FLOPs (preferred) or a validated FLOPs model that matches your architecture and token counts.

Show that your main scaling fits are stable when you change training budget (more epochs or fixed step counts) and early-stop consistently.

Make the overfitting threshold criterion objective and show the exponent is stable under reasonable definitions.

Add uncertainty for the loss to rejection mapping and validate it across sizes and feature sets.

Clean the references and fix incorrect attributions.


Questions for the authors?

Why is one epoch the right budget for the scaling-law surface fit? What changes if you train longer and early-stop?

Can you report results with measured FLOPs, or justify your compute proxy against a real FLOPs estimate for this Transformer?

How sensitive is the overfitting threshold exponent to your labeling rule?

Does the loss to QCD rejection mapping hold across model sizes and input configurations without refitting? Show fit quality and uncertainty.

---

### Meta-Review · Area_Chair_rfEv · 2026-02-27

**Recommendation:** Accept (Poster)
**Confidence:** 4

**Metareview:**

This submission has received two reviews. One reviewer rated this submission as "clear accept" and another reviewer rated this work as "Ok but not good enough"

After reading the reviews and weighing the strengths and weaknesses as reported by both reviewers, I recommend this paper for "acceptance" and ask the authors to include the feedback of both reviews into the camera ready version of the paper.

---

### Decision · Program_Chairs · 2026-03-03

Accept (Poster)